# Application of Physical and Chemical Enhanced Backwashing to Reduce Membrane Fouling in the Water Treatment Process Using Ceramic Membranes

**DOI:** 10.3390/membranes8040110

**Published:** 2018-11-15

**Authors:** Seogyeong Park, Joon-Seok Kang, Jeong Jun Lee, Thi-Kim-Quyen Vo, Han-Seung Kim

**Affiliations:** Department of Environmental Engineering and Energy, Myongji University, 116 Myongji-ro, Cheoin-gu, Yongin-si, Gyeonggi-do 17058, Korea; joonseok0724@gmail.com (J.-S.K.); dlwjdwns87@naver.com (J.J.L.); vokimquyen77@gmail.com (T.-K.-Q.V.)

**Keywords:** ceramic membrane, chemical enhanced backwashing (CEB), membrane recovery, steam cleaning

## Abstract

This study investigated the improvement of operating efficiency through physical cleaning and chemical enhanced backwashing (CEB) using ceramic membranes with high permeability and chemical safety compared to organic membranes. The turbidity and DOC (Dissolved Organic Carbon) concentrations were selected to ensure that the degree of contamination was always constant. The operating pressures were fixed at 100, 200, and 300 kPa, and the filtration was terminated when the effluent flow rate decreased to 30% or less from the initial value. After filtration, backwashing was performed at a pressure of 500 kPa using 500 mL backwash water. The membrane was cleaned by dipping in NaOCl, and a new washing technique was proposed for steam washing. In this study, we investigated the recovery rate of membranes by selectively performing physical cleaning and CEB by changing the influent water quality and operating pressure conditions.

## 1. Introduction

As people’s standard of living has improved, interest in quality water resources has increased. This has led to the development of advanced water treatment processes using membranes. The membrane separation process, which has advantages such as simplification of facilities, minimization of site area, and stable water quality through automation, shows excellent efficiency compared with the conventional sand filtration process [1,2]. However, the membrane filtration process causes fouling in which particles and organic matters dissolved in the water are deposited on the membrane surface. Thus, there are some operational problems, such as increasing the transmembrane pressure (TMP) and reducing the quantity of the permeate. The two major cleaning methods are physical and chemical cleaning. Physical cleaning uses only clean water and air. Chemical cleaning can be described as CIP (cleaning in place) and CEB (chemical enhanced backwashing) when classified with respect to washing cycle and chemical concentration. CIP is used to control contamination that cannot be controlled by CEB cleaning with a lower concentration of chemicals. CIP requires a high concentration of chemicals and takes a long time; consequently, CIP causes problems such as low operation efficiency and high treatment cost [3,4].

Steam carries high thermal energy. The bond between membrane and materials is broken when the thermal energy of steam is higher than the bond energy of the materials making up the membrane. This phenomenon is called as thermolysis. Steam cleaning separates particles and dissolves organic matters deposited on the membrane surface by transferring thermal energy, which is considered to be able to control the fouling. Steam cleaning can be compared with the conventional physical and chemical cleaning method.

Commercialized membranes are mainly made of organic polymers. Recently, inorganic membranes have gained more attention due to characteristics of inorganic materials, which include being resistant to high temperature, high concentrations of chemicals, and organic solvents. Therefore, inorganic membranes, especially ceramic membranes, can be operated under extreme conditions, such as high temperature, high pressure, and widely ranging pH values. The characteristics of ceramic membranes are shown in Table 1 [5].

In the case of an organic membrane in a general water treatment process, Fe and Mn dissolved by the acid used in the chemical cleaning are deposited on the surface of organic membrane and generate biofouling. On the other hand, ceramic membranes are highly resistant to chemicals when they are chemically cleaned with an acid or a base, so there are few restrictions on chemical cleaning. In addition, they are suitable for steam cleaning applications, since they can be operated under extreme high temperature conditions [5].

The purpose of this study is to improve the operational efficiency of a water purification facility through physical cleaning and CEB using an inorganic membrane. In addition, we will develop a cleaning method by applying steam without using a high concentration of chemicals. It can be used in industrial fields which require thorough hygiene management, such as the food industry, because of its environmentally friendly characteristics [6].

## 2. Materials and Methods

### 2.1. Ceramic Membrane Filtration Process

Ceramic membrane filtration experiments were performed using the apparatus shown in Figure 1. Filtration was carried out at various pressures of 100, 200, and 300 kPa.

The raw water characteristics were set based on the average turbidity and dissolved organic carbon (DOC) concentration of the raw water of the Seoul water treatment plant in Case A. In addition, high-turbidity conditions and high DOC concentration were set in Cases B and C. The characteristics are shown in Table 2 [7].

Kaolin (SHOWA chemical, Gyoda, Japan) and humic acid (SIGMA ALDRICH, St. Louis, MO, USA) were used to define turbidity and organic matter, respectively. Humic acid was used after filtration using a 1.2 µm glass microfiber filter (GF/C 0.47, Whatman, Maidstone, UK).

In this experiment, the flux measurement was carried out by measuring the mass of the treated water in real time and converting mass into a volume. DI (Deionized water) filtration was carried out before and after raw water filtration to measure the flux before and after raw water filtration. In addition, the flux reduction rate was calculated by flux measurement during filtration. After cleaning, the flux was measured using the same method to confirm the membrane recovery rate. For the ceramic membrane device, one MF membrane of alumina material from METAWATER Co. (nominal pore size 0.1 µm, membrane surface area 0.035 m^2^) was used. Detailed specifications of the membrane and a photograph of the membrane are shown in Table 3. The temperature, UV_254_, DOC and turbidity of influent and treated water were measured. Before UV and DOC measurement, the sample was filtered using a 0.45 µm PVDF syringe filter (GD/X syringe filter, Whatman, Maidstone, UK). The analyzer for each measurement category is shown in Table 4.

### 2.2. Method of Washing

After the membrane filtration process, the fouled membrane was cleaned using the following procedures.

#### 2.2.1. Physical Cleaning

DI and air were injected perpendicularly onto the membrane surface at 500 kPa to clean the fouled membrane. The entire process was carried out for about 30 s, and the washing was carried out with a pressure time of 20 s, water cleaning time of 3 s, air cleaning time of 1 s, and a pressure relief time of 5 s.

#### 2.2.2. Chemical Cleaning

Chemical cleaning was set at 300 ppm NaOCl, which is 10% of 3000 ppm, the concentration of the chemical used in the CIP. The cleaning was carried out by dipping the fouled membrane for 10, 20, 30 min and 1, 2, 3, 4, 5, 6 h [8].

#### 2.2.3. Steam Cleaning

As shown in Figure 1, the steam cleaning is intended to reduce the bonding force between the contaminants by allowing high-temperature steam to pass through the membrane. The saturated steam temperature was 120 °C, and the injection pressure was 100 kPa. Steam injection was performed for up to 3 min. After steam injection, physical cleaning and DI filtration were performed to measure the flux.

### 2.3. Evaluation of Cleaning Recovery

#### 2.3.1. J_T_/J_0_ and V_T_/V_A_

There is a difference in flux depending on operating pressure conditions. Therefore, to compare the rate of change of the flux, J_T_/J_0_ was estimated, J_T_: real-time flux during filtration, J_0_: flux during DI filtration before raw water filtration. Also, the amount of raw water that can be flowed depends on the characteristics of the raw water. Therefore, V_T_/V_A_ was estimated, V_T_: Volume of permeate during raw water filtration at certain time, V_A_: volume of water permeated during raw water filtration when the filtration was terminated. The comparison of the characteristics of the raw water was carried out using V_T_/V_A_.

#### 2.3.2. Membrane Recovery Rate

To evaluate the washing efficiency, the DI was filtered before and after filtering the raw water. The weight of the treated water was measured and recorded in units of 10 s. In addition, the flux was calculated using the membrane area and the weight of the treated water. The flux at the end of the raw water filtration shows the degree of reduction of the flux by filtration and the flux during DI filtration after cleaning represents the extent to which the cleaning has recovered the membrane.
$$\mathrm{Membrane}\;\mathrm{recovery}\;\mathrm{rate}=\frac{J_{2}-J_{1}}{J_{0}-J_{1}}$$

J_0_: flux during DI filtration before raw water filtrationJ_1_: flux at the end of raw water filtrationJ_2_: flux during DI filtration after raw water filtration

## 3. Results and Discussion

### 3.1. Comparison of Membrane Recovery Rate after Physical and Chemical Cleaning

NaOCl used in chemical cleaning is effective in cases where the main cause of membrane contamination is organic matter. Thus, it was performed by using raw water Case C, with 10 NTU, 8 ppm, for comparison of the chemical and physical cleaning. In the case of chemical cleaning, the fouled membrane was immersed for 10, 20, 30 min, 1, 2, 3, 4, 5, 6 h, and then DI filtration was carried out to calculate flux. Figure 2 shows the membrane recovery rate with time of chemical cleaning for different pressures. Flux reduction rates after the filtration process were in the range of 90–94% at all raw water and pressure conditions. At each pressure condition, the membrane recovery rate was 20% lower than the recovery rate for 6 h chemical cleaning, with values of 25%, 17.65% and 18.18%, respectively. However, in the case of 100 kPa, the membrane recovery rates were 72.5% and 94.5%, respectively, after 30 min of chemical cleaning. After 2 h of chemical cleaning, a membrane recovery rate of 100% was observed. The membrane recovery rate in 200 kPa was also increased, and the rate reached 86.3% after 6 h of washing. In the case of 300 kPa, the membrane recovery rate showed a steady increase similar to that seen for 200 kPa. However, the amount of increase was low, at 3.6–7.2%, with a membrane recovery rate of 33.5% after 6 h of cleaning time. On the other hand, when the recovery rate by chemical cleaning and physical cleaning were compared under the various respective operating pressures, the physical cleaning recovery rates were similar to the chemical cleaning recovery rate for 3 h, at 43.1% and 18.6% at 200 and 300 kPa, respectively. In the case of 100 kPa, the membrane recovery rate was particularly high for chemical cleaning, and exceeded the physical cleaning recovery rate before the chemical cleaning of 30 min.

### 3.2. Difference of Contamination According to Raw Water Characteristics

Figure 3 shows J_T_/J_0_ variation patterns according to each pressure when all the conditions are the same without raw water characteristics. Under all pressure conditions, J_T_/J_0_, which used 10NTU, DOC 8 ppm high-organic-matter water, decreased to 20.9–26.3% with an initial ratio of effluent of 20%. When the filtration was completed, the ratio of flux was 8.7%, 3.6% and 2.7% for each pressure. J_T_/J_0_ using 10 NTU, DOC 8 ppm high-organic-matter water was low at the end of filtration compared to the J_T_/J_0_ values of other raw water, which were 25.0–35.7%, 14.3–21.4%, and 11.4–12.7%. Therefore, it was confirmed that the high-organic-matter raw water generated severe contamination at the initial stage during filtration with the ceramic membrane, thereby greatly reducing the permeation rate of the membrane. 

At 100 and 200 kPa, the high-turbidity-water of 25 NTU, DOC 2.5 ppm, and 10 NTU and DOC 2.5 ppm raw water showed similar patterns, and J_T_/J_0_ decreased. In the case of 300 kPa, J_T_/J_0_ of raw water of 10 NTU and 2.5 ppm decreased with a steep slope compared to values at an operating pressure of 100 and 200 kPa. At the end of filtration, J_T_/J_0_ of 10 NTU, DOC 2.5 ppm was 12.8%, which was 22.9% lower than the J_T_/J_0_ value of 35.7% at the end of filtration for 100 kPa. In the case of high-turbidity water of 25 NTU and DOC 2.5 ppm, J_T_/J_0_ decreased sharply at the initial stage and decreased to 15.9% at 75% of V_T_/V_A_. J_T_/J_0_ at the end of filtration was 11.4%, and later, J_T_/J_0_ was 75%, confirming that a low volume of treated water was yielded due to fouling by high-turbidity water filtration.

The membrane recovery rates by physical cleaning and J_T_/J_0_ at the end of filtration are shown in Table 5. Comparing the two different levels of turbidity in the raw water, the membrane recovery rates were 66%, 38%, 32% for 10 NTU raw water, whereas the recovery rates for high-turbidity raw water of 25 NTU were 22%, 19%, 12%, respectively. Therefore, fouling by high-turbidity water is seen to be an irreversible contamination which is difficult to recover by physical cleaning. J_T_/J_0_ at the end of filtration of high-organic-matter water of DOC 8 ppm was remarkably low, at 2.7–8.7%, and the initial contamination occurred rapidly. However, the membrane recovery rates of physical cleaning were 50.0%, 43.1%, and 18.6%, which were somewhat lower than those for 10 NTU, and it was confirmed that the proportion of reversible contamination that could be recovered by physical cleaning was higher than that for 25 NTU and DOC 2.5 ppm.

### 3.3. Application of Steam Cleaning

To confirm the feasibility of steam cleaning as a new technique for membrane cleaning, we compared the results of single physical cleaning and steam cleaning, which injects steam onto the fouled membrane before physical cleaning using organic membranes commercialized in the water treatment process. A high-turbidity water of 25 NTU, DOC 2.5 ppm was used to generate rapid contamination at the lab scale. The filtration was carried out under a constant-pressure operation condition of 100 kPa. In addition, DI filtration at the end of raw water filtration decreased by 94–95% with respect to that before filtration. The steam injection time was set to 1, 2, and 3 min. A single physical cleaning, without steam injection, showed a low membrane recovery rate of 1.3%. When steam cleaning was carried out, the membrane recovery rates were 45.6%, 44.7% and 43.8% for each steam injection time, and the membrane recovery rate was more than 40%, regardless of the steam injection time. Therefore, it is confirmed that the steam cleaning process is effective for recovering from membrane contamination.

The steam cleaning process was applied to the ceramic membrane due to the fact that the steam cleaning process was effective in increasing the membrane recovery rate of the organic membrane. Ceramic membranes are considered a suitable target for adding a steam cleaning process because of their high resistance to heat and chemicals.

Mid-turbidity organic-matter water of 10 NTU, DOC 2.5 ppm, high-turbidity water of 25 NTU, DOC 2.5 ppm, and high-organic-matter water of 10 NTU, DOC 8 ppm were measured with regard to flux. In addition, using the measured flux, the filtration flux reduction rate and the recovery rate by physical cleaning and steam cleaning were derived. The results are shown in Figure 4. The higher the pressure, the lower the membrane recovery rate by physical cleaning. In the case of high-turbidity water, the membrane recovery rate of physical cleaning was remarkably lower under all pressure conditions. When steam injection was carried out for 3 min using 10 NTU and DOC 2.5 ppm raw water, increases in the membrane recovery rate of 11.1%, 19% and 14.7% were observed, respectively. When the raw water of 25 NTU, DOC 2.5 ppm was filtered, the increase in the membrane recovery rate with the addition of 3 minutes of steam injection was 34.6–53.8%. This is about a three times higher membrane recovery rate than the physical cleaning membrane recovery rate. Raw water of 10 NTU, DOC 8 ppm also showed an increased membrane recovery rate of 16.7–22.2%. Therefore, it is concluded that the addition of the steam cleaning step was effective in controlling the fouling of the ceramic membrane.

### 3.4. The Role of Steam Washing

The addition of a steam injection led to an increase in membrane recovery rate under all raw water conditions. Therefore, the range and possibility of steam cleaning were confirmed through a comparison of the use of steam cleaning and chemical cleaning for membrane maintenance cleaning, and the results are shown in Figure 5. The results in Figure 5 show that the membrane recovery rate of chemical cleaning decreased sharply as the operating pressure increased. However, steam cleaning showed a stable efficiency of 65% or more in 100 and 200 kPa. At 300 kPa, steam cleaning showed a low membrane recovery rate of 36.8% compared with 100 and 200 kPa, but 3.3% higher than the membrane recovery rate of 6 h of chemical cleaning. In case of high-pressure conditions, which cause severe organic pollution, it is considered that the steam cleaning effect is higher than the chemical cleaning.

## 4. Conclusions

In this study, the changes in flux and recovery rate of ceramic membranes were compared after various cleaning methods, including a new steam cleaning technique, physical cleaning, and chemical cleaning. The higher the operating pressure, the lower the membrane recovery rate appeared to be under all conditions. When the operating pressure was low, membrane recovery by chemical cleaning was predominant. However, as the operating pressure increased, membrane recovery by chemical cleaning decreased sharply. In addition, it was confirmed that the membrane recovery by steam injection for 3 min exceeded the recovery rate of chemical cleaning for 6 h. It is clear that the steam cleaning is effective for controlling the organic pollution generated at a rapid rate. From the evaluation of the cleaning ability by the steam cleaning by measuring the membrane recovery rate, steam cleaning can be recommended as a primary cleaning process in order to enhance the control of fouling and lowering the frequency of chemical cleaning.

## Figures and Tables

**Figure 1 membranes-08-00110-f001:**
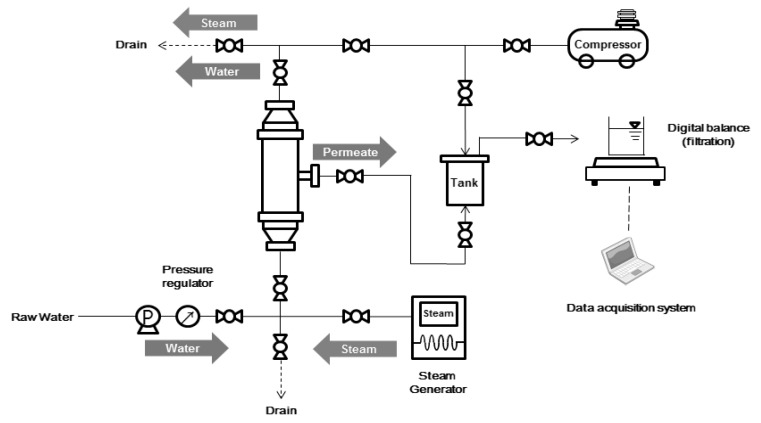
Schematic diagram of a lab-scale system.

**Figure 2 membranes-08-00110-f002:**
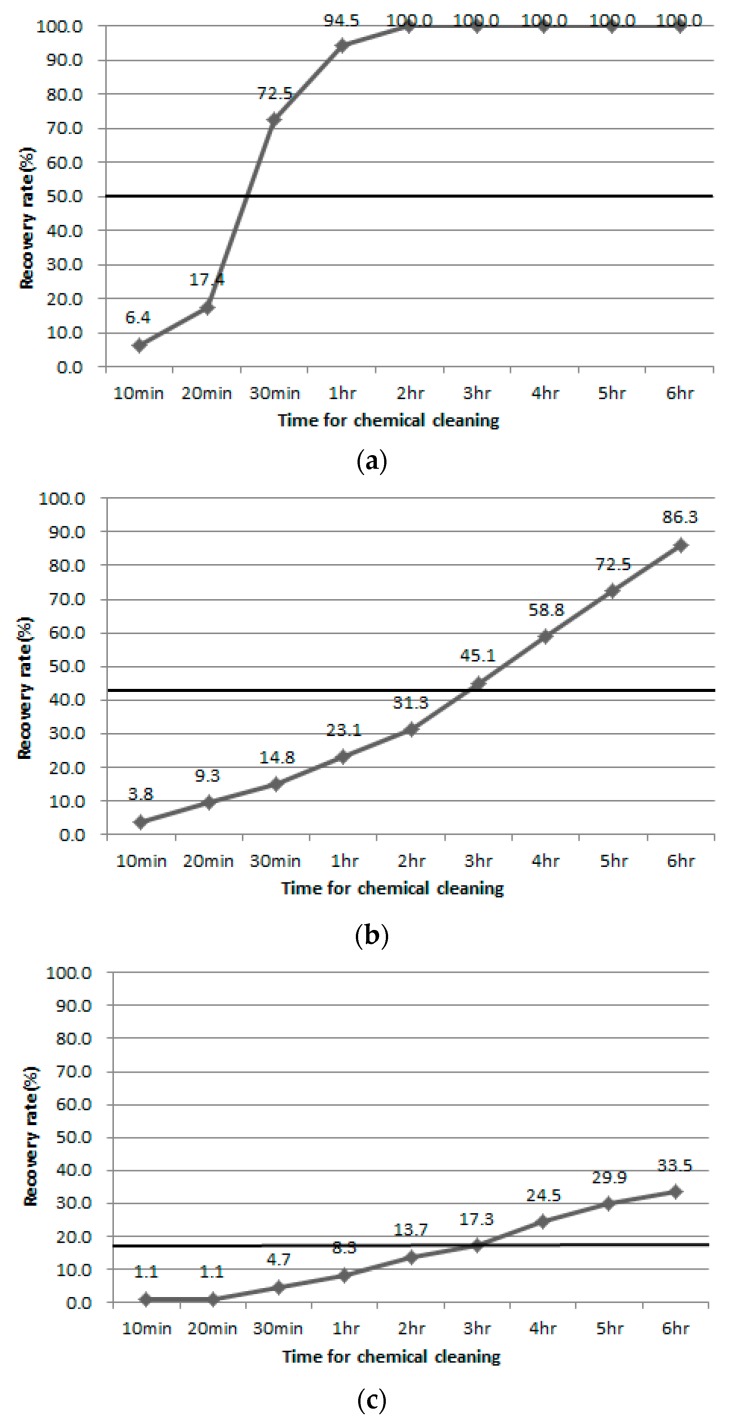
Membrane recovery rate after chemical cleaning at (**a**) 100 kPa, (**b**) 200 kPa, (**c**) 300 kPa.

**Figure 3 membranes-08-00110-f003:**
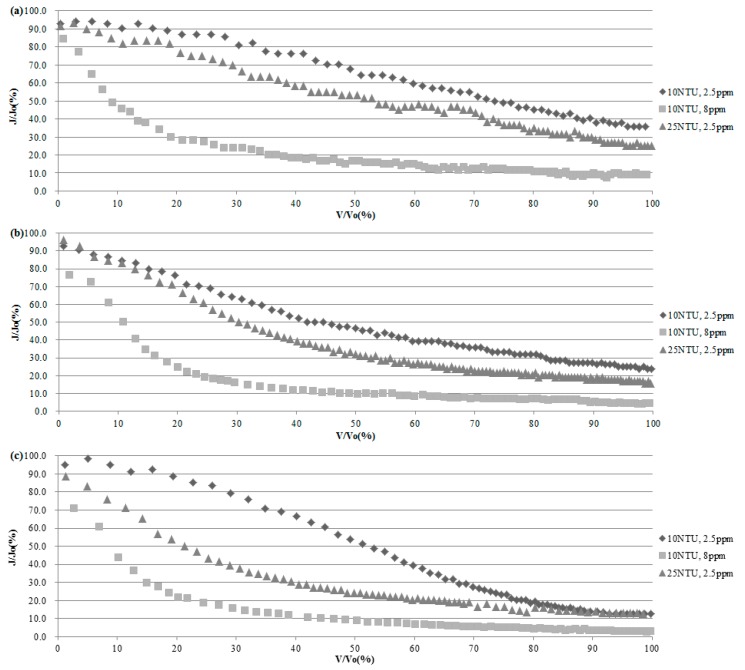
Change of J_T_/J_0_ according to V_T_/V_A_ in (**a**) 100 kPa, (**b**) 200 kPa, (**c**) 300 kPa.

**Figure 4 membranes-08-00110-f004:**
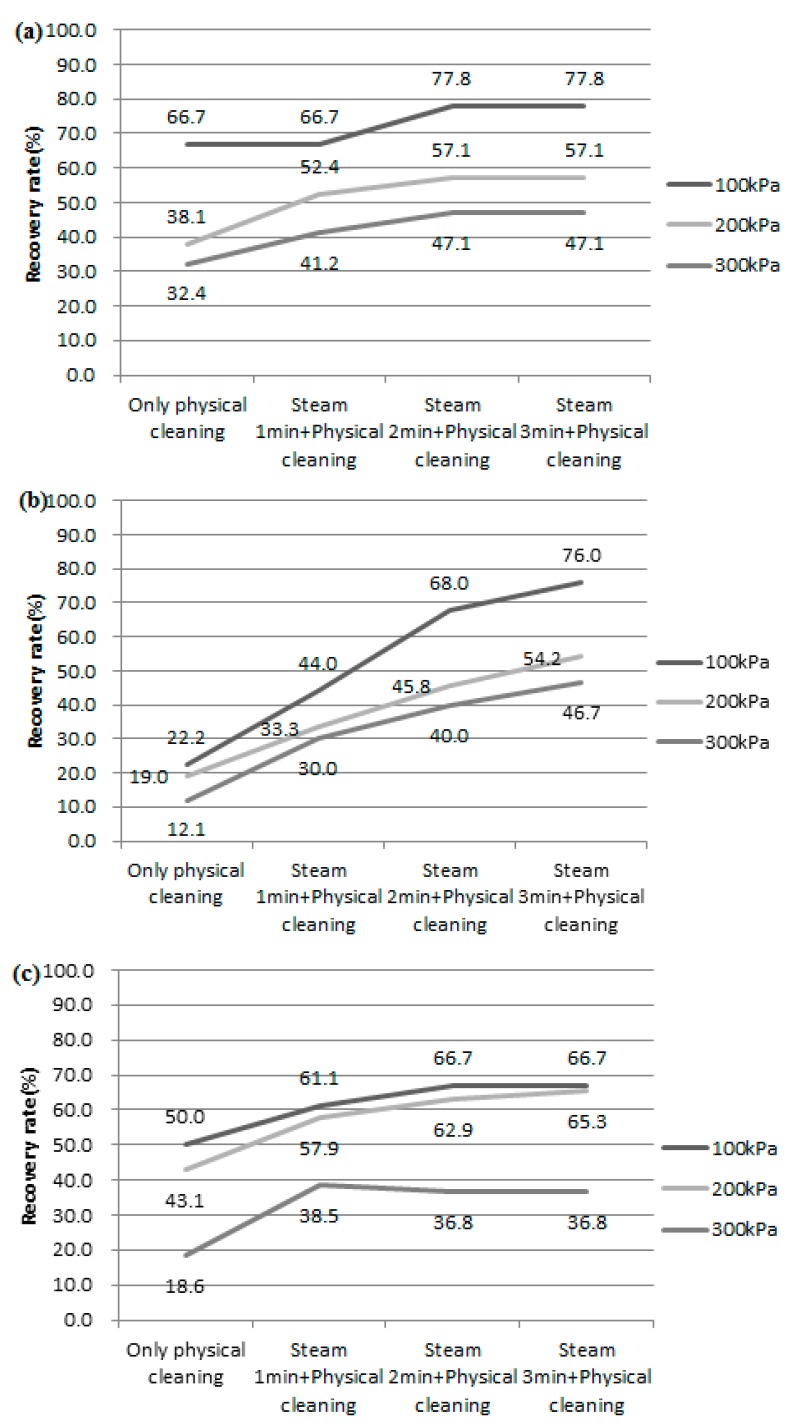
The membrane recovery rate changes with steam injection added to the physical cleaning using raw water of (**a**) 10 NTU, DOC 2.5 ppm, (**b**) 25 NTU, DOC 2.5 ppm, (**c**) 10 NTU, DOC 8 ppm.

**Figure 5 membranes-08-00110-f005:**
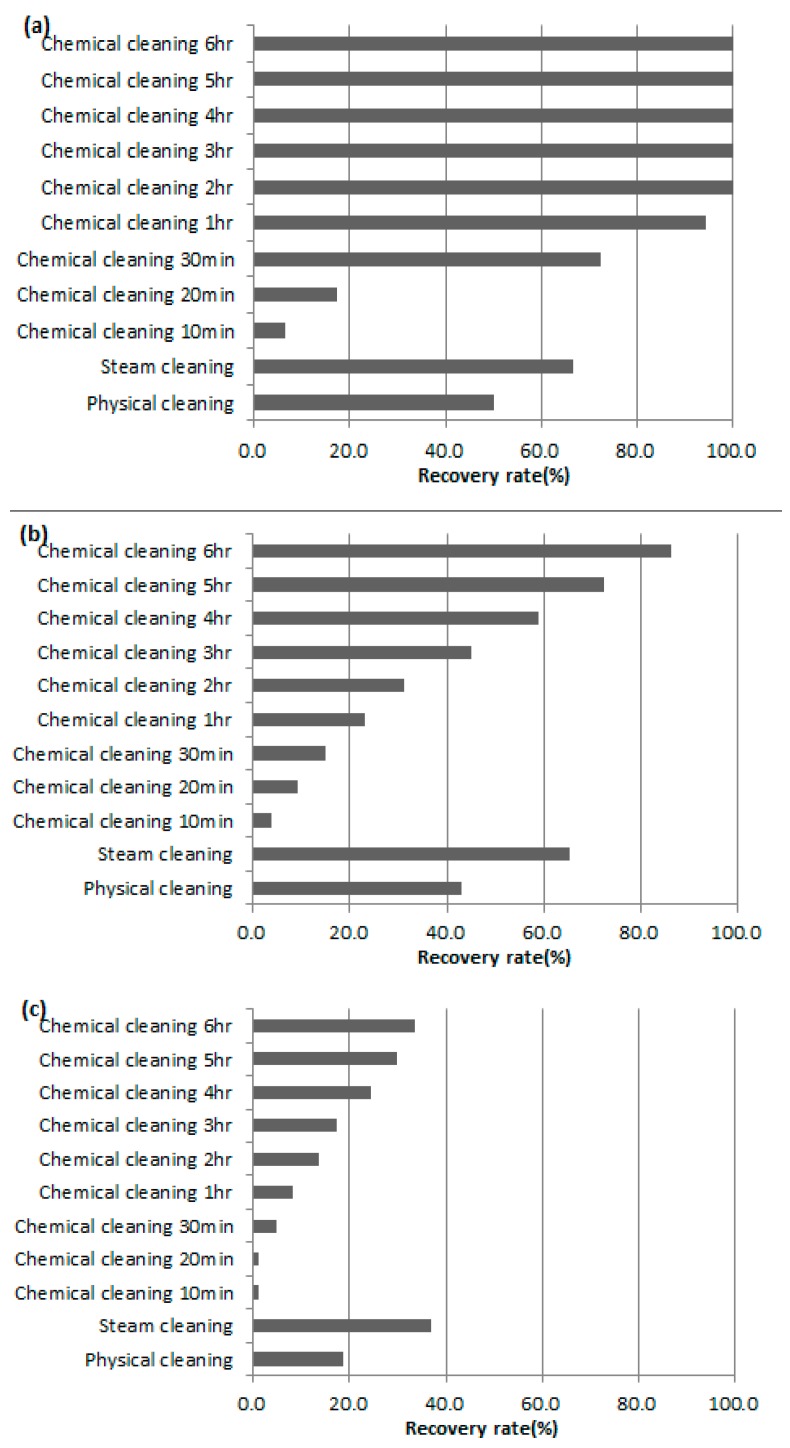
Comparison of membrane recovery rate of chemical cleaning and steam cleaning in (**a**) 100 kPa, (**b**) 200 kPa, (**c**) 300 kPa.

**Table 1 membranes-08-00110-t001:** Characteristics of ceramic membrane [5].

Advantages	Disadvantages
High-temperature thermal stability	High cost: 5 times higher than polymer membranes
High chemical stability: High corrosion resistance	
Long-term use	Easy to break due to high brittleness
High mechanical strength	

**Table 2 membranes-08-00110-t002:** Characteristics of raw water.

Parameter	Case A	Case B	Case C
Turbidity	10 NTU	25 NTU	10 NTU
DOC concentration	2.5 ppm	2.5 ppm	8 ppm

**Table 3 membranes-08-00110-t003:** Characteristics of ceramic membrane.

Categories	Contents	
Membrane Type	MF ceramic membrane	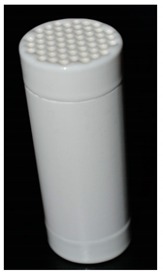
Material	Ceramic (Al_2_O_3_)	
Type	Inner-pressure monolith	
Nominal pore size	0.1 µm	
Membrane surface area	0.035 m^2^	
pH range	0~14	
Max. Operating pressure	20 kgf/cm^2^	
Manufactory	METAWATER (Tokyo, Japan)	

**Table 4 membranes-08-00110-t004:** Analytical instruments and method.

Categories	Analyzers	Etc.
Turbidity	2100N Turbidimeter, HACH (Loveland, CO, USA)	NTU (Nephelometric Turbidity Unit)
UV_254_	UV-1800, Shimadzu (Kyoto, Japan)	Ultraviolet photometer
DOC	TOC-V_CPH_, Shimadzu (Kyoto, Japan)	NPOC (Non-purgeable organic carbon)
Temperature	Orion 3star, Thermo (Walthan, MA, USA)	Degrees celsius

**Table 5 membranes-08-00110-t005:** Physical cleaning recovery rate and the J_T_/J_0_ at the end of filtration.

Parameter	10 NTU, DOC 2.5 ppm			10 NTU, DOC 8 ppm			25 NTU, DOC 2.5 ppm		
	100 kPa	200 kPa	300 kPa	100 kPa	200 kPa	300 kPa	100 kPa	200 kPa	300 kPa
Recovery rate of physical cleaning (%)	66.7	38.1	32.4	50.0	43.1	18.6	22.2	19.0	12.1
J_T_/J_0_ at the end of filtration (%)	35.7	25.0	12.8	8.7	3.6	2.7	25.0	14.3	11.4

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
