# Peer review of "Application of Physical and Chemical Enhanced Backwashing to Reduce Membrane Fouling in the Water Treatment Process Using Ceramic Membranes"

_membranes, 2018, doi:10.3390/membranes8040110_

Round 1

Reviewer 1 Report

The manuscript is focused on the investigation of how physical cleaning and chemical enhanced backwashing can improve operating efficiency.

The presented manuscript is prepared in an appropriate way, however, there is a lack of novelty. This topic is well known in the scientific literature. Furthermore, the introduction section needs to be improved. Real state of art should be presented, referring more scientific papers suitable to the topic of the work.

Experimental section, please present data about producers of chemicals/equipment in a consistent way. Furthermore, the presented method should be described more in detail, e.g. section 2.2.

The manuscript needs to be corrected from the linguistic point of view.

Please organize the manuscript in a more careful way, e.g. please refer description to Fig. 2. Now there is missing. And please correct description on page 5. Presented data in the text are different then the values presented in Fig. 2.

Lines 130-133

“However, in the case of 100kPa, the membrane recovery rate was 72.5% and 94.5%, respectively after 30min of the chemical washing, and after 2 hours of chemical washing, 100% of the membrane recovery was observed. Membrane recovery rate in 200kPa was also increased and the rate reached 86.3% in 6 hours of washing.”

According to the data presented in Fig. 2, it should be in the following way:

100kPa, the membrane recovery rate was 14.8% and 31.3%, respectively after 30min of the chemical washing, and after 2 hours of chemical washing, 100% (these are data for 200kPa, not 100kPa!!!) of the membrane recovery was observed. Membrane recovery rate in 200kPa was also increased and the rate reached to 86.3% (here 100% or these are data for 100kPa) in 6 hours of washing.

Generally, all sections need to be well organized.

Reference should be prepared in a uniform way.

In the presented form, the manuscript is not ready to be published. 

Reviewer 2 Report

After revision I think that this work could be suitable for publication in Membranes journal. My recommendation, after the paper revision, is: minor revision.

My detail comments:

1. The title is too long

2. text lines 217-226 shoud be before Figures 5

3. line 220 - add to: shown in Fig 5.

4. The authors can be compare their results to the results of other researchers

Reviewer 3 Report

The work is devoted to a very important aspect of membrane filtration – reduction of membrane fouling. The results on chemical and steam washing of ceramic membranes are shown and discussed. However, the manuscript before publication should be substantially improved.

Here are some detailed comments.

The Authors use shortcuts which are not defined; e.g.:

page 1: CIP, CEB; p.3: “… by filtering the DI before and after…”.

Page 3: “Detailed specifications of the membrane and photographs of the modules are shown in the Table 3.” – only one photo of one module is shown.

The description “2.2.1. Physical washing” is not clear: “DI water and air are injected at 500 kPa to wash the membrane. The entire process is carried out for about 1 minute, and the washing is carried out with a pressure time of 20 sec, a water washing time of 3 sec, an air washing time of 1 sec, and a pressure relief time of 5 sec.” – 20+3+1+5 = 29 s, not 60 s.

P.4: “The washing was carried out by dipping the membrane for a certain time.” – what time?

Subsections “2.3.1. J/Jo and V/Vo” and “2.3.2. Membrane recovery rate” – it is better to express the formulae for these quantities in terms of short symbols and to supply their full description. The notation should be logical.

“The effluent volume and flux ..” – what flux?

“Figure 2. Membrane recovery rate after immersion in the chemicals, which NaOCl, (a)100kPa, (b)200kPa, (c)300kPa.” – the caption should be completed: versus what it is plotted? Insert a space between value and unit.

p.5: “In the case of chemical washing, the flux was measured by DI filtration every 10, 20, 30 min, 1, 2, 3, 4, 5, 6 hr.” – what does mean that time?

“…in the case of 100kPa, the membrane recovery rate was 72.5% and 94.5%, respectively after 30min of the chemical washing, and after 2 hours of chemical washing, 100% of the membrane recovery was observed.” – no such values are seen in Fig. 2a.

Here I’ve stopped reading the manuscript.

Please, check and correct your manuscript very carefully before re-sending it to the Editor.

I’m not a native English speaker but I feel that the language should be substantially improved; e.g.:

page 2: “However, interest in ceramic membranes, which are inorganic membranes, has increased since Japan.”;

 “300 kPa for same amout of volume water”;

p.3: “…the flux was measured in the same method …”;

“…from Meta Water Co.(Nominal pore size 0.1 , Membrane surface area 0.035 m2) ...” – lack of space, capital letters;

“… in the Table…” – “the” can be removed;

“When UV and DOC were measured, the sample was filtered by 0.45 PVDF Syringe filter (GD/X Syringe filter, Whatman) and then used.” - ??

p.4: “The chemical washing of the CEB …” = “The chemical washing of the chemical enhanced backwashing” – it does not sound good;

“…ratio of flux and effluent volume were calculated …” – ratios;

“At this time, Flux was recorded in units of 10 seconds by calculation by measuring the weight …” - ??

p.5: “… the flux was measured by DI filtration …” - ?

Round 2

Reviewer 1 Report

The manuscript has been corrected in an appropriate way. Authors took into consideration comments and suggestions. In the received version manuscript can be published.